# Emergence of power and complexity in obstetric teamwork

**Christopher Neuhaus**[1]*, **Dag Erik Lutnæs**[2], **Johan Bergström**[3]

**1** Department of Anesthesiology, University Hospital Heidelberg, Heidelberg, Germany, **2** Ukom, Norwegian Healthcare Investigation Board, Stavanger, Norway, **3** Division of Risk Management and Societal Safety, Lund University, Lund, Sweden

* c.neuhaus@uni-heidelberg.de

## Abstract

### Background

Recently, increasing attention has been paid to team processes in peripartum care settings with the aim to improve fetomaternal outcomes. However, we have yet to understand how the perception of teamwork in peripartum care is shaped in a complex, multi-disciplinary environment.

### Methods

The aim of this study was to approach the question using qualitative social-scientific methodology. The theoretical foundation of the study was that obstetric teamwork is the result of a balancing act in which multiple goal conflicts are continuously negotiated and managed right at the boundary of acceptable performance in a complex adaptive system. We explored this theory by gathering lived experiences of successful management of peripartum emergencies. Based on our analysis we generated an understanding of teamwork as a phenomenon emerging from interpersonal relationships, complex relations of power, and the enactment of current quality management practices.

### Results

Caregivers define teamwork through the quality of their collaboration, defined by respect and appreciation, open communication, role distribution, and shared experiences. However, teamwork also becomes the framework for negotiation of many conflicts that originated elsewhere. Power was the core theme that emerged in the analysis of our participants' narratives, which is in stark contrast to the otherwise promoted egalitarian rhetoric of team training. While our participants generally reverted to explanations based on their professional identities, traditions or cultures, interesting dynamics become visible when work is viewed through the power lens.

### Conclusions

Our study paints the convoluted picture of a work environment with all its intricacies, constraints, interpersonal relations and hierarchical struggles that are much more

**Data Availability Statement:** Data cannot be shared publicly because of privacy considerations of the study participants and interview content that can potentially identify subjects despite anonymisation. Data are available from the Ethics Committee of the Ruprecht-Karls-University

Heidelberg (contact via ethikkommission-I@med.
uni-heidelberg.de) for researchers who meet the
criteria for access to confidential data.

**Funding:** The author(s) received no specific
funding for this work.

**Competing interests:** The authors have declared
that no competing interests exist.

**Abbreviations:** CRM, Crew Resource
Management; ATA, Applied Thematic Analysis; OB,
Obstetrician; MW, Midwife; ANA,
Anaesthesiologist; SD, Standard deviation; CTG,
Cardiotocography; BE, Base excess; IQTiG,
Institute for Quality Assurance and Transparency in
Healthcare.

representative of a complex system rather than the easily tractable environment that so
many stakeholders would like healthcare practitioners to believe in. The issue of power
emerged as a decisive factor in the social dynamics at the workplace, revealing hidden
agendas in the teamwork discourse.

## Introduction

The dynamic nature of Obstetrics, the risks involved, and the need for constant adaptation on
part of the care providers inevitably lead to increased interest in team processes involved in the
provision of peripartum care [1]. As the physiological event of giving birth is similar all over
the world, it is probably due to this 'common physiologic ground' that different traditions,
myths, taboos and practices are socially constructed around childbirth [2]. While often a
"medical/social-model of childbirth" [3] is dichotomized as the contrast between obstetrical
and midwifery practice, reality usually resembles a much more nuanced negotiation in
between: A complex web of interpersonal, inter-professional and hierarchical relationships
presents in obstetric care [4], and success and resilience as a result of this diversity can be con-
sidered a 'petri dish' for further examination of team processes.

Current approaches to teamwork in obstetrics resemble the predominant behavioristic,
normative approach generally found in medicine [5], both regarding the scientific basis used
for the argument and the discursive language. A trend of simplification can be traced, albeit
often disguised by professional pragmatism. As an example, one can consider the use of the
concept of Situational Awareness [6] in an obstetric context: "Situational awareness is a con-
cept that was first defined in aviation. In obstetrics, it has been difficult to define and reliably
measure [. . .]; *however, put simply, it refers to knowing what is going on*" [7]. This is in stark
contrast to ongoing debates about the general viability of the concept in human factors
research [8–10]. Moreover, simplicity is explicitly desired: "In order to achieve better outcomes
[. . .], it is essential that team-training interventions are *simple and relevant* to the maternity
care setting" [7]. Nonetheless, one can also find a critical discourse around team training in
obstetrics, where crew resource management (CRM)-based interventions have failed to dem-
onstrate the desired improvements [11]. Furthermore, in a review of teamwork performance
measurement tools in obstetrics, Fransen et al. [12] evaluated six different frameworks avail-
able for simulated settings and noted both limited evidence for their psychometric properties
as well as the general lack of a "gold standard" for teamwork performance. This mimics other
critical reviews on performance measurements in healthcare [13, 14]. We question whether
this current approach to teach and 'improve' teamwork addresses the needs of practitioners in
dynamic, resource-constrained and goal-conflicted work environments [15].

The knowledge about teamwork in obstetrics remains ambiguous and conflicted. 'First, do
no harm' remains the guiding moral imperative for frontline staff to constantly reflect upon
and improve their work practices in an effort to increase fetomaternal safety in the perinatal
setting. The relevant questions then become both what constitutes, and what shapes the care-
giver's perception of, 'good teamwork'? The aim of this study was to explore how successful
teamwork is constructed and perceived by those directly involved in patient care and contrast
these findings with traditional normative approaches. We strive to contribute to the debate
using a qualitative social-scientific approach to teamwork, and explore the theory that obstetric
teamwork is the result of a balancing act in which multiple goal conflicts are continuously
negotiated and managed right at the boundary of acceptable performance [16] in a complex
adaptive system. For this purpose, multiple perspectives of successful management of peripar-
tum emergencies are gathered in the setting of a German university hospital.

## Methods

We conducted an exploratory case-study [17] at a large University Hospital in Germany. Participants engaged in a semi-structured face-to-face interview following a pre-approved interview guide (see Table 1 and S1 Appendix).

Interview language was German. The interviews were audio-recorded, and recordings were later transcribed using f4transcript for Mac® (dr. dresing & pehl GmbH, Marburg, Germany) and translated into English. All personal or identifying information was removed during the transcription process. Data was analyzed using Applied Thematic Analysis (ATA). Drawing from a multitude of theoretical and methodological perspectives, its "[. . .] primary concern is with presenting the stories and experiences voiced by study participants as accurately and comprehensively as possible" [18]. Guest, MacQueen [18] describe data analysis in ATA as "locating meaning in the data". Distinct to the often-encountered idea of sensemaking in qualitative research, this reinforces a measured approach that is cautious of highly imaginative over-interpretation of problematic data. Our analytic strategy was therefore designed according to recommendations by Guest, MacQueen [18] with the main goal of providing an "audit trail" of the process rather than an analytic "black box" that leaves many questions up to the reader's imagination (for a complete description of the methodology, see S2 Appendix and the codebook in S3 Appendix). Quantitative data analysis is restricted to descriptive analysis of the participants (occupation, experience) and the interview process (number of interviews, average length) and was performed using Microsoft Excel for Mac 2011 ® (Microsoft, Redmond, WA, USA). All qualitative data analysis, as well as data visualization, note taking, recording of memos and analysis ideas was performed using NVIVO 12 for Mac ® (QSR International, Melbourne, Australia). The iterative process of data analysis consisted of the following phases:

1. *Establishing clear analytic objectives*: The main purpose of our study was to harness the narratives of practitioners and develop an understanding for their conceptualization of teamwork. This is highly exploratory in nature.

2. *Data quality control and enhancement during data collection*: During the interview process, the two researchers frequently exchanged experiences and compared notes regarding the interviews to control the quality of data gathered and ensure reliability, and to already familiarize oneself with the data and develop a 'feel' for recurring themes and ideas voiced by the participants.

3. *Text segmentation and quality control*: After transcription, interview data was segmented by questions as specified in the interview guide. This was later used to assess and compare the consistency of the questions asked and to provide a foundation for comparability. This segmentation was purely structural in nature and meant to support our methodological approach. We were especially conscious of the ongoing controversy regarding text segmentation and its potential for distorting context and meaning [19], therefore the original interview dialogue was preserved and used for all coding and analysis of content.

**Table 1. Interview questions.**

| |
|---|
| **1. First, try to think of a time (the last time?) where you experienced a peripartum emergency where the work was successful. Using your own words, please tell me about it.** |
| **2. What, in your mind, made it successful?** |
| **3. What makes work successful in general?** |
| **4. Consider a colleague that you perceive as good and successful in working together with others, which qualities makes you put them in high regard?** |
| **5. In your opinion, is there a correlation between good work and good outcome for mother and child?** |

4. *Development of initial themes and codes*: In the initial mapping of our data, both researchers together looked for representations of teamwork aspects as described by Manser [20, see Table 2] in the participant's narratives, while at the same time generating emergent themes and defining vague boundaries around them.

5. *Development of a codebook and content coding*: In the next iteration, themes were more closely defined, restructured, and a codebook developed for definite coding of the text (see S3 Appendix). Subsequently, all interviews were coded for content by both researchers together. This process was repeated twice, where the codebook was further refined, and analysis notes, ideas and memos were recorded for later analysis. It is important to note at this point that steps 3–5, while easily broken down into a logical sequence of events, can hardly represent distinct processes. Guest, MacQueen [18] note that "[...] the act of identifying a meaningful segment of text calls for some minimal representation of that meaning as a code, a note, a query, or a tag". Subsequently, coding ideas come to mind even during the initial reading and structural segmentation of the text. On the other hand, codes that were conceived and hardly ever applied were later changed, or redefined, as the process (r) evolved.

6. *First-order data analysis and comparison*: Subsequently, we analyzed data to provide a description of how teamwork was constructed by our participants. Also, quantification methods were used, mostly to visualize code frequencies and aid the researchers in pattern recognition.

7. *Second-order analysis*: During the second-order analysis, the sometimes incomplete, sketchy or contradictory data is connected with theoretical literature with the aim of constructing theoretical explanations. According to Shkedi [21], the purpose of these explanations is not necessarily the construction of "grand theory", but rather a conversion and organization of the content and the connection with the researcher, as it is impossible to separate the inquirer from the inquired from a constructivist epistemological perspective. For us, this not only meant making sense of the analysis, but aligning the results with our own experience and understanding of the delivery of peripartum care.

8. *Writing the report*: Rather than a mere form of representation, writing "[...] plays an active part in the process of organizing, working with and analyzing data" [19]. It is also a way of crystallizing vague conceptions and connecting loosely formulated ideas, and presenting them to the reader through sound reasoning and exemplification.

The study was conducted in accordance with the Declaration of Helsinki and was approved by the Ethics Review Committee of the Medical Faculty of the Ruprecht-Karls-University

**Table 2. Participants' roles and experience.**

| No. of participants | | 13 |
|---|---|---|
| **Roles** | Obstetrician | 5 |
| | Midwife | 6 |
| | Anesthesiologist | 2 |
| **Experience** | 0–2 years | 1 |
| | 3–5 years | 3 |
| | 6–10 years | 7 |
| | 11–20 years | 2 |

Heidelberg (Ref: S-110/2018). The manuscript adheres to applicable EQUATOR guidelines (See S2 Appendix).

## Participants and demographics

Participants were recruited using convenience sampling as a type of non-probabilistic sampling [22]. Following a three-step procedure, first the target population was determined by the inclusion criteria specifying that participants had to be aged 18 years or older and belong to one of the following professions: Board-certified obstetrician or anesthesiologist, certified or registered nurse or midwife. In a second step, the sample frame was defined by the study site that this project was designed around. Third, individual enrolment was based on subject availability and accessibility, with a target sample size of 10–15 participants. The sample size was in part based on an estimated degree of data saturation, but also based on pragmatic considerations regarding the researchers' time for as well as the scope of this project. Participants engaged in a semi-structured interview with one of the primary investigators. The interviews were conducted face-to-face, nobody except the participant and the researcher was present during the interviews.

A total of 13 healthcare professionals were interviewed between June and November 2018. Table 2 provides an overview of participants' roles and experience.

Mean interview duration was 29.6 minutes (SD 6.4 min.). All interviews showed the structure specified in the interview guide. As the interview questions #1–3 built on one another, on two occasions participants already intuitively gave answers to subsequent questions during the conversation, which led the interviewer to not explicitly ask these questions again. All instances were reviewed by the research team, which concluded that the interview content followed the logic of the pre-structured script and that structural reliability was not compromised. Quotes are referenced by profession (OB–obstetrician, MW–midwife, ANA–anesthesiologist) and participant number. In the transcriptions, [. . .] denotes that a part of the quote was left out or changed to preserve context or anonymity, while . . . signifies a pause on the recording.

## Validity & reliability

In addition to the aforementioned audit trail, we took the following steps to increase the validity of our research during various stages of the project:

The study design, including the format and questions of the semi-structured interview, was developed by both principal researchers together. This ensured familiarity with the research objectives and the envisioned methods for data acquisition.

The interviews were conducted by the same researchers that designed the study, therefore the purpose of each question was known as prerequisite for further inductive probing. During the interview phase, we constantly monitored and compared data as it was collected, so probing techniques could be aligned and overall data consistency enhanced.

Transcriptions were made to provide verbatim accounts of the data collection event, including the captioning of reactions (thought phases, pauses, laughs, sighs, etc.). In addition to providing more precise accounts of the interviews than mere field notes, we were able to capture and document rich, powerful and sometimes very emotional quotes that convey much more meaning than any analysis ever will.

A codebook was developed through multiple iterations. Also, coding was conducted together, so that ambiguities and reliability problems due to different interpretations could be immediately resolved.

## Results

### What is good teamwork?

When analyzed by existing frameworks that describe relevant aspects of teamwork in health-care [20], the most frequently addressed theme concerned the "quality of collaboration". For participants, this aggregate construct is defined by the degree of mutual respect and support, trust, and reliability. This was especially relevant for midwives when describing their daily collaboration on the ward. Mutual respect was evoked as prerequisite for the clear definition of boundaries. The concepts of "teamwork" and "team play" appeared heavily interwoven for the participants:

*If you notice that the other one doesn't have time for something you take over. But. . . you also know your limits, how far you can go [. . .] For example among midwives that's the case with vaginal examinations, that's up to the midwife that. . .well that takes care of that woman. And if I do a CTG [Cardiotocography] and I see that it doesn't look good I won't simply examine her [. . .] but notify her midwife [. . .]. Don't just do it. It's about communication and team play. (MW5)*

An important factor mentioned by participants, apart from a professional collaboration, was the quality of interpersonal relationships as basis for all further teamwork considerations.

*I think, generally speaking everybody has more fun and things work better if you get along, and it's like you feel that if you get along and treat each other nicely you tend to take over jobs that are primarily not your own, I can certainly say that [the midwives] put in IVs for me and draw blood if it is stressful and they see me running around and uhm. . .of course there are those where you don't get along, and they really stick to where they simply don't do things. [. . .] I think uhm in reality it's important that you see everything as a team, that you make progress as a team, what you can do even if it's theoretically not your job, you can still do it. (OB4)*

This directly influences not only the quality of collaboration, but also how team members communicate:

*Well if the atmosphere is bad, everybody tries to avoid each other somehow, and then that gets to be a time issue I believe, uhm or time gets to be a relevant issue because something is communicated only if it's really bad at the end somehow, like an extremely bad CTG uhm. . .and the threshold to call somebody at night, if everybody is [expletive, read annoyed], is of course. . .is higher than if you have a more or less collegial relationship. (OB2)*

Furthermore, interwoven with aspects of interpersonal relations and the quality of collaboration are shared experiences as common ground for collaboration. The participants' descriptions of how teamwork is affected based on trust and knowledge of others in dynamic situations show how their experiences defy behavioristic approaches from CRM-like training concepts:

*I guess you can train the basics, where you actively try to listen to the other, but for that to be really fruitful it depends on how well you can judge the other person. I'm sure there were a few things that I did at night, where I spontaneously delivered kids together with experienced midwifes, that probably wouldn't have worked at a different time with maybe young*

*inexperienced midwifes and inexperienced residents, simply because. . .well you know that you can get involved in a riskier situation together. (OB5)*

On the surface, good teamwork is constructed as a combination of the quality of collaboration, defined by respect and appreciation, open communication, role distribution, and shared experiences.

## Emergent phenomena

Over the course of our interviews, different themes began to emerge that seem to greatly influence and model teamwork in obstetrics. Most notably, these concerned individual personality, role perception, and hierarchy. We began to form an understanding of teamwork as an emergent property of how work is organized, structured and performed by individual actors, as opposed to a mere set of behavioral properties that are learned and applied. It became increasingly apparent that separate realms existed for midwives and obstetricians, based on role perception, organizational structures and hierarchy. In the midwives' narratives, their primary area of responsibility concerned taking care of "their" women during the process of giving birth, sometimes for hours, forming a deep personal bond in the process. References to teamwork primarily related to collaboration with fellow midwives, and only in a secondary step involved obstetricians, or physicians from other disciplines. Moreover, there's a clear distinction between physiologic and pathologic deliveries, depending on the degree of medical intervention required:

*Everyone agrees as long as the delivery is physiological it's in the midwife's hands, and if the delivery or the postpartum phase turns pathologic it is the doctor's job. (OB5)*

This distinction turned out to have far-reaching consequences for the concept of teamwork. Rather than constituting a shared understanding of how work is organized and enacted together, team composition, properties, and roles depend on this classification, and fundamentally shift once it is changed. Subsequently, responsibility as a property of taking care of the delivery, is usually not shared, but passed on between groups in a process that can be explicit or implicit. This is further complicated by the fact that, oftentimes, young inexperienced residents work together with experienced midwives. As one resident summarized, all this results in a complex work environment where formal and informal hierarchies need to be constantly negotiated by those working together:

*Well the immediate partner of the obstetrician is the midwife, in a way that. . .it's the bigger challenge to handle that together. And on the other hand, and some midwifes say that openly, of course it's like they don't need us at all, and it's true. That's how it's always portrayed. . .it's something natural, in 95% of cases it's not a risky situation, and that's true, but in the end it comes down to formalities, if the maternity ward is led by physicians the doctor has to be present at the end of the delivery, but some of course spin it in a way where they say "we really don't need you, don't do anything, just stand there, we'll do the delivery, and you can do the suturing afterwards". And that's a little bit. . .you get the feeling that some have a wrong understanding of one's role. But well, that's how it is. It's surely dependent on how you see your role as a midwife, or on the setting. [. . .] Of course, in most cases there's nothing, you just stand there as a doctor and watch and in the end all is well. And in my experience, it's good if you don't. . .involve yourself unnecessarily, because the midwife is the primary contact person for the mother, and she knows how. . .she's known her for hours, and I, well. . .with most mothers I haven't even shaken their hand, I just join them, say hello and then the child is*

*delivered, therefore it's always like. . .the midwife has been there for maybe even 12 or 8 hours and knows about the situation and. . .but of course, if something's wrong in the end the responsibility is transferred to the physician, and well. . .you have to see that clearly, the midwives tend to forget that the doctor ultimately carries the responsibility, and he's the one who faces the heat if something's wrong, they like to forget that. (OB2)*

As our curiosity was sparked, further inquiry into 'facing the heat' revealed a culture of outcome-based accountability among obstetricians that would later prove pivotal to our understanding of how teamwork is constructed.

*[G]enerally you have to explain the pH in the morning, then the CTG is looked at and. . .it's a shame, because the CTG doesn't reflect the situation, you know, there's so much going on, [the mother] is screaming, you can't talk to the patient, uhm, back and forth, uhm, she's in pain uhm and the CTG is recording permanently and the only thing that's communicated in the morning is that CTG and two pH values. . .it's a little bit like we condense everything down to those values, it has to be quantifiable, but it's like. . .a little bit like. . .(sighs) that's the reason why everybody is so fixated on formalities, that everything looks good on the outside, you know, uhm, that everything is kept below the surface, and then. . .and that we don't get problems in the morning. (OB2; similar statements made by other obstetricians in this study)*

This aspect of accountability, which is not shared by the team, turned out to be a major factor in obstetric decision making, especially when considering prolonged natural birth versus elective caesarean section, a source of much disagreement between obstetricians and midwives. Young residents, when faced with difficult decisions, tend to 'play it safe', not wanting to end up on the wrong end of an inquiry, while midwives focused on the experience of giving birth perceive parts of the process as unjustified medical intervention.

*I think that we pathologize a lot, I can't deny that. That's normal. Because [the doctors] are lacking the basics, the understanding of what's normal. (MW3)*

From the midwives' point of view, the lack of accountability is not only acknowledged, but even sometimes welcome.

*And like I said I am glad that I can say here's the point where I am allowed to hand things off [. . .] Where I can say "this is not my job anymore, that's over now and I don't want to be responsible for this". (MW1)*

The aspect of accountability, while partly acknowledged by the midwives, turns into questions of trust within the team.

*And that's what [the doctors] are afraid of, I think. To trust the midwife if they know that [if it goes wrong] the next morning their boss will give them [a hard time]. (MW3)*

In summary, pursuing questions of accountability proved revelatory of more profound ontological differences. As a midwife explains:

*Look, we learn differently. In our case, well I don't know but I'm guessing in med school you learn maybe. . . during all of med school there's maybe four semesters of physiology, max, and the rest is just pathologies. Maybe I'm exaggerating, but physiology plays a smaller part I*

*would say, and you'll learn everything that is pathologic and what you have to do. In our case it's the complete opposite, there's two years of physiology and they only talk about pathologies in your final year. What you need to do if it doesn't work the way you imagined. So, you know in our case the design is different, and that's why we approach things more positively, because first we see the natural, and we get to know the natural better [. . .]. (MW2)*

## Discussion

When viewed from a more meta perspective, many aspects of our participants' narratives that seemed confusing and contradictory at first revealed two logical paradoxes.

1. Although glad to be able to hand off responsibility, midwives often voice notions of not feeling valued or appreciated, especially if those responsible disagree with a proposed assessment or treatment option. They feel conflicted about the involvement of additional medical professionals in the delivery process; while generally perceived as outsiders, they might also represent the required resource to avoid adverse outcomes and increase the systems' overall adaptive capacity.

2. Clinical work is organized in a way that physicians, while formally responsible, are occupied with a multitude of tasks, and can therefore only be superficially involved in the delivery process. At the same time, their performance is not judged with considerations of their local rationality [23], which would take into account what made sense to them at the time, but measured according to outcome-based quality indicators. To deal with these idiosyncrasies, obstetricians not seldomly revert to a "play-it-safe" approach to decision making that defeats any advantages from an integrative, interprofessional, team-based approach. In an exemplary fashion, this shows how the quality agenda is effectively able to hijack safety efforts. This is obfuscated by an accompanying rhetoric constantly appealing to the individual actors' belief that "good teamwork" will benefit the patient and solve any potential problems.

Consequently, teamwork becomes the framework within which many conflicts are negotiated that, in reality, originated elsewhere, e.g. the respective hierarchies or goal conflicts within the involved professions. This is further confounded by two separate dimensions of hierarchy: While physicians are hierarchically superior to midwifes in a medical sense, there is no normative connection between physicians' and midwives' hierarchies in an organizational, disciplinary context. The implications for teamwork under these circumstances become increasingly visible when the notion of power is considered.

### Power in obstetric teamwork

It has been previously noted that a representation of organizational life without the consideration of power may result in serious shortcomings [24]. Power, in its various dimensions, was the core theme that emerged in the analysis of our participants' narratives, which is in stark contrast to the otherwise promoted egalitarian rhetoric of team training. While our participants generally reverted to explanations based on different professional identities, traditions or cultures, interesting dynamics become visible when their work is viewed through the power lens.

Antonsen [24] introduces a three-dimensional view on power based on Lukes' [25] classification of power, supplemented by descriptions on possible sources of power in organizations:

1. In its most obvious and observable form, the first dimension of power is visible in behavior and overt conflict. This type of power, also characterized as 'power over', may be rooted in

position, knowledge and expertise, control of rewards and resources, coercive power, or individual attributes like charisma, energy or political skills.

2. The second dimension of power, summarized as covert conflict, is expressed through non-decisions and the ability to control decision-making processes in organizations. It often concerns *potential issues*, which are prevented from becoming overt issues through non-decision making and the withholding of information. It is important to note that this is not contradictory, but rather represent a complementary dimension of power to the first.

3. The third dimension of power is concerned with political agendas and latent conflict, and examines how meaning is constructed in social life. Lukes notes that "[. . .] power is at its most effective when least observable" [25]. Consequently, the third dimension of power describes a more systemic way of how power is exercised by one entity over another without being noticed, often through mechanisms of socialization and obfuscation. Although a certain overlap exists between the three dimensions, power as exercised in the third dimension expands our understanding to include strategic considerations that enable sociological rather than merely personalized explanations.

When exploring the notion of power as embedded in obstetric teamwork, it is important to base this on the understanding that there are never-ending disagreements about how the concept of power is to be defined, employed, and studied. According to the author, "[. . .] how much power you see in the social world and where you locate it depends on how you conceive of it, and these disagreements are in part moral and political, and inescapably so." [25].

Power, in its most obvious form, is easily identified in our participant's narratives: Obstetricians have the authority, and thus power, to direct the course of a delivery by indicating medical interventions, e.g. a Caesarean section. Coercive power is executed by superiors when residents have to justify their decisions in hindsight during morning rounds. In more general terms, this dimension of power is perceived as a positive, or driving, force behind successful teamwork. However, it is not tied to hierarchical status, as midwives have the power to involve physicians from a variety of domains, and transfer authority, thus shedding themselves of formal responsibility.

Subtler, the second and third dimensions of power are represented in covert and latent conflict between obstetricians and midwives. This is not only reinforced, but created in the first place by a system that differentiates between formal hierarchy and experience, and places the former in the hands of junior obstetricians on the journey of acquiring the latter. It becomes apparent whenever decisions have to be made by residents against the will of midwives, which hold much informal power through their knowledge and obstetric experience. One resident explains the situation where a caesarean section was delayed, against the resident's better medical judgement, due to this covert power struggle, resulting in an unwanted fetal outcome:

> *It's been a few years ago, uhm, during one of my first shifts on call. . .I probably, well formally I surely. . .would have had to insist, [. . .] and I, well I simply didn't exude the authority, and didn't reinforce that in communication, I have to admit, I didn't go in the team room and crash the handover and say come on guys [. . .] we are going NOW. And of course, that's what the boss was all over me for the next morning. (OB2)*

The resident also offers a glimpse into the complexity of social interactions and the constant negotiation between normative and epistemic power taking place at the workplace. Epistemic power can be defined as the ability to influence what others believe, think, or know, as well as the ability to enable and disable others from exerting epistemic influence [26]. According to

this distinction, while especially young residents hold, and even try to exercise, normative power, this is potentially counteracted by epistemic power on part of the midwives. Interestingly, these processes rarely fit into frameworks and formal descriptions of how work should best be performed, or organized.

Another source of much informal (and epistemic) power lies in information about the course of the delivery, and the power to decide on when to involve the obstetricians on part of the midwives. Although this interaction has been targeted through a multitude of approaches, be it "best practice", guidelines, or team training, such approaches are at the heart of what Dekker, Bergström [4] characterize as essentially meaningless normative rhetorical commitments ignoring the complexity of obstetrics.

> If the problems associated with obstetric intervention were merely complicated, the solution would lie in optimizing, through best practice guidelines, the intervention criteria, and sensitivity to evidence of those closest to the obstetric process. But a complex system cannot be reduced to the behavior or compliance of individual components. It is about understanding the intricate web of relationships they weave, their interconnections and cross-dependencies, and the constantly changing nature of these as people come and go and technologies get adapted in use. [4]

To connect the aspect of complexity to power and teamwork, it is important to remember the high emphasis our participants placed on relationships and personal knowledge when describing successful teamwork, a factor where proponents of standardization and team training would strongly disagree as to its importance. The aforementioned discrepancy between formal disciplinary and medical hierarchies underscores how more subtle mechanisms are evoked to govern interprofessional relations whenever the limits of normative hierarchical ones are reached. Again, teamwork rhetoric provides the framework for much organizational negotiation, while more dormant issues like resident training and workload, or discussions about suitable quality indicators and their connection to departmental funding remain below the surface.

## Quality vs. safety

In an exemplary fashion our study into teamwork revealed how the quality-management agenda can effectively influence clinical teamwork in unforeseen ways. In 2015, Germany founded the Institute for Quality Assurance and Transparency in Healthcare (IQTiG) with the aim of evaluating, overseeing and assuring healthcare quality. Although an independent, academic institution, it has a federal mandate to develop instruments that measure healthcare quality and enable the transparent comparison of different hospitals [27]. A standard of care is defined through a set of quality indicators, and hospitals are required to provide associated data that are compared using statistical instruments. Deviations from the norm result in official inquiries with possible implications for future hospital certification and funding [28].

For obstetrics, the quality indicators include the APGAR score after 5 minutes, as well as Base Excess (BE) and pH-values of the umbilical cord as surrogates for fetal depression and distress. Repeatedly, our participants described how clinical decision making was influenced by the requirement to monitor, report and justify these parameters. Midwives not seldomly scoff at doctors' decisions to perform caesarean sections, writing them off as pre-emptive, based on an exaggerated need for safety and indication of a lack of traditional obstetric experience. Furthermore even the quality of teamwork is affected, as midwives often see their own competency undervalued and questioned when their suggestions to continue a natural delivery

are overruled in favor of a caesarean section. On the other hand, the reasoning of junior doctors having to "face the heat" in the morning is equally understandable, especially with the consideration of organizational and federal peculiarities.

While these repercussions were certainly not intended when federal requirements for quality reporting were introduced, it shows the (perhaps unintended) consequences of quality management operationalized as an effort to standardize and quantify, thereby ignoring qualitative, interpersonal and social intricacies of a complex workplace. The delivery of care ultimately comes down to the one-on-one interaction of humans that will always require a degree of compassion and dedication that cannot be automated or standardized. It would almost amount to cynicism to state that the system that relies on these properties being exerted by its practitioners, is unable to extend to them the same amount of individual consideration in return. It also reinforces the intractable quality of healthcare as a complex adaptive system [29].

## Limitations

Our study has several limitations. The overall scope of the project establishes clear boundaries in terms of generalizability of our findings. We were always aware that it is an exploratory case-study that might prove to be hypothesis-generating at best, and descriptive of local, insulated and circumstantial phenomena at least. However, using additional data sources and triangulation, we try to provide some frame of reference for the reader to put our findings into perspective. This is also the reason why we try to harness another source of potential bias, our own domain knowledge in healthcare and medical simulation, and for one of us (CN), our continuing involvement in the active management of peripartum emergencies as part of our professional duties.

Interviewing study participants, no matter how well scripted, will elicit biased responses, based on a multitude of factors. All study participants were at least vaguely acquainted with the investigator, and some had previously worked with him on several occasions. When reading through the interviews, however, the honest, critical and open responses to our questions can be seen as testament to the participants' professionalism, and to the atmosphere and setting the interviews were conducted in. Due to logistical constraints, data collection methods complimentary to individual interviews (e.g. focus groups) could not be performed.

During qualitative data analyses, it was helpful for the process of reflexivity that the other researcher (DEL) was impartial, and the joint analysis and coding could help reveal and deal with biased interpretation. This is also one reason why we chose to do all the coding and analysis work together, thereby accepting the fact that we would not be able to calculate a Cohen's Kappa as formal inter-rater reliability score, but had to revert to a mere subjective assessment. Also, our analysis work was not subject to external review.

## Conclusion

Instead of providing easy answers to our research question, our participants' narratives paint the convoluted picture of a work environment with all its intricacies, constraints, interpersonal relations and hierarchical struggles that are much more representative of a complex system rather than the easily tractable environment that so many stakeholders would like healthcare practitioners to believe in. The issue of power emerged as a decisive factor in the social dynamics at the workplace, revealing hidden agendas in the teamwork discourse. We could show how ultimately, local team processes can become influenced by the pursuit of system-wide quality agendas in a tightly-coupled, complex system. This should lead us to question the kind of support and training obstetric teams require to perform their everyday jobs of providing

safe peripartum care, despite sometimes adversarial structural, organizational, logistical or political conditions.

## Supporting information

**S1 Appendix. Interview guide.**
(DOCX)

**S2 Appendix. Methodology and audit trail.**
(DOCX)

**S3 Appendix. Codebook.**
(DOCX)

## Author Contributions

**Conceptualization:** Christopher Neuhaus, Dag Erik Lutnæs, Johan Bergström.

**Data curation:** Christopher Neuhaus, Dag Erik Lutnæs.

**Formal analysis:** Christopher Neuhaus, Dag Erik Lutnæs.

**Investigation:** Christopher Neuhaus, Dag Erik Lutnæs.

**Methodology:** Christopher Neuhaus, Dag Erik Lutnæs.

**Project administration:** Christopher Neuhaus.

**Supervision:** Johan Bergström.

**Validation:** Johan Bergström.

**Writing – original draft:** Christopher Neuhaus, Dag Erik Lutnæs.

**Writing – review & editing:** Johan Bergström.

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
