## [Decision Letter · Decision Letter 0]

23 Feb 2022

PONE-D-21-32827Emergence of Power and Complexity in Obstetric Teamwork: an observational Study.PLOS ONE

Dear Dr. Neuhaus,

Thank you for submitting your manuscript to PLOS ONE. After careful consideration, we feel that it has merit but does not fully meet PLOS ONE’s publication criteria as it currently stands. Therefore, we invite you to submit a revised version of the manuscript that addresses the points raised during the review process.

We look forward to receiving your revised manuscript.

Kind regards,

Dylan A Mordaunt, MB ChB, MPH, MHLM, FRACP, FAIDH

Academic Editor

PLOS ONE

Journal Requirements:

Additional Editor Comments:

Thank you for your submission. There are detailed responses from the reviewers below. With regards to the criteria for publication:

1. The study appears to present the results of original research.

2. Results reported do not appear to have been published elsewhere.

3. Experiments, statistics, and other analyses are performed to a reasonable technical standard- the reviewers have identified gaps and issues that should be addresed.

4. Conclusions are presented in an appropriate fashion and are supported by the data.

5. The article is presented in an intelligible fashion and is written in standard English.

6. The research meets all applicable standards for the ethics of experimentation and research integrity.

7. The article adheres to appropriate reporting guidelines and community standards for data availability.

Reviewers' comments:

Reviewer's Responses to Questions

**Comments to the Author**

1. Is the manuscript technically sound, and do the data support the conclusions?

Reviewer #1: Partly

Reviewer #2: Partly

Reviewer #3: Partly

2. Has the statistical analysis been performed appropriately and rigorously? 

Reviewer #1: N/A

Reviewer #2: No

Reviewer #3: N/A

3. Have the authors made all data underlying the findings in their manuscript fully available?

Reviewer #1: No

Reviewer #2: No

Reviewer #3: No

4. Is the manuscript presented in an intelligible fashion and written in standard English?

Reviewer #1: No

Reviewer #2: Yes

Reviewer #3: Yes

5. Review Comments to the Author

Reviewer #1: Dear Researchers,

I was interested to read this from a methods point of view. I am not an expert in this area and provided a detailed review on methods leaving other sections to other reviewers. I felt currently that I wasn’t clear on the methods and some improvement was needed. I have identified this below. I wish you the best going forward.

Abstract

Methods

Please identify the paradigmatic stance and methodology selected.

Please identify inclusion criteria

Please consider methods used to obtain data, analysis approach undertaken.

Methods main text

Appendix B seems to have more information, one concern for me was the volume of expected information here within the text e.g., there is no sampling technique named here, no consideration to sample size and why on the numbers, no consideration to a section on rigour or quality – you may mention this in appendix B but it is no good there.

Line 101 – you call the work here a case study, but your title is an observational study? You also perform thematic analysis which in the results doesn’t seem to pull out information as case studies may? Consider clarifying this for the reader

Line 101 – you use reference 17 to name your approach a qualitative case study – however looking at this reference I could only find limited references to case study approaches. Can you identify unique characteristics of this methodology or give a better reference because a reader wont be able to understand the methodological process you are required to consider? Please remember you need to link your paradigm with methodology and quality criteria e.g., an interpretivists criteria could vary hugely compared to someone who assumes a critical realist criteria. I note you mention validity and reliability in your Appendix B this may be ok in some paradigms views but consideration is needed

Line 101 – do you consider a checklist like CARE or equivalent to aid the information here?

Line 102 – what do you mean by a pre-approved interview schedule? How did you decide on the questions? Does it link to literature? Did you pilot this first and if so what changes were made? Was there any PPI involved?

You refer to Appendix A and within it cite Mansers work that splits the concept of TEAM work up – how does Table 1 in the text link to Table 1 in the appendix?

Analysis section follows – however for me the 8 stages of analysis in Appendix B can be named here and Appendix B needs to illustrate the application and examples of each stage as applied.

Results

Clarity over type of methodology used is needed

Consideration to minor themes within major themes could be considered.

Reviewer #2: PRISMA FLOW DIAGRAM shows that the no. of records after duplication is 83. However, in the next stage, the record screened (52) and records excluded (32) indicates a total of 84 records, which is not matching with the earlier stage (n=83).

Details of components used for quality assessment of the included studies are not described.

The authors should mention the value of I-square (usually >25%) for deciding the heterogeneity level.

Figure 3 is a funnel plot, and it is not a forest plot as mentioned.

Further, in the funnel plot, it is clearly shown that all the prevalence values are outside the 95% confidence limits, indicating a clear publication bias. However, the authors stated that there is no publication bias, which is not true.

Since there is a publication bias, the p-value of the egger test should be significant (p<0.05). Therefore, the authors should carefully interpret the results.

Sample size & author (Figure 5) is not required.

Figures 7 & 8 are shown with eight studies; the reason for the inclusion of one more study is unclear.

In the Page 13, the statement, “The possibility of women’s discontinuation of implanon for those women lacking counselling during service delivery were 2.54 times (OR: 2.34, 95%CI: 2.98, 2.77) more likely compared with women getting adequate counselling” is wrong due to wrong presentation of OR.

In the Page 13, the last sentence is stated as, “The possibility of discontinuation of implanon among women who lack the satisfaction of provided service was 4.42 times (OR: 4.42, 96% CI: 2.73, 7.15) more likely than their counterparts”. Why did the authors suddenly adopt 96% CI for this OR?

Meta-regression is a part of meta-analysis to determine the significant factors contributing to heterogeneity. Even though there is a high heterogeneity in the study, the authors did not carry out meta-regression to assess significant factors contributing to heterogeneity.

Limitation of study needs to be mentioned

PRISMA checklist 2020 is missing

Reviewer #3: When I was invited to review this manuscript it was especially the phrase “… we have yet to understand how the perception of teamwork … is shaped in a complex, multi-disciplinary environment” that raised my interest, as I thought I could relate to my own prior work on systems science around issues of dynamic complexity in the delivery of health care. But let me make a disclaimer: I’m not an obstetrician (not a clinician even) and neither do I have a background in andragogy or socio-anthropology.

My comments are as follows.

• Since this was an exploratory study I find it strange to see the Introduction section being concluded with the sentence “Based on our analysis we generate an understanding of teamwork as a phenomenon emerging from interpersonal relationships, complex relations of power, and the enactment of current quality management practices.” This begs the question whether the researchers really had an open mind in this study – or in other words: I’m a little suspicious that this study was conducted in this particular university hospital for a certain reason, possibly because of an issue (conflict?) that had come to the surface earlier – and if so, the authors might have had some experiences (preconceived ideas?) that guided this study in a particular direction, i.e. that of power imbalances among the members of the obstetrical team. I am not against an N=1 study, but I do feel that an explanation is required why the study was conducted at this particular hospital.

• I am missing a study objective!

• The methods section claims that “the theoretical foundation of the study was that obstetric teamwork is the result of a balancing act in which multiple goal conflicts are continuously negotiated and managed right at the boundary of acceptable performance in a complex adaptive system”. While I agree that (obstetric) team work has multiple features, many of which may mutually influence each other, I’m not sure whether the term ‘multi-disciplinary’ applies. More importantly, however, a conceptual framework around teamwork dynamics from sectors other than obstetrics – or even from sectors outside healthcare – might have informed the study design. It appears that the five interview questions (which seem pretty general) and the fairly short duration of the interviews (30 minutes on average) did not allow for much depth.

• The sentence “We explored this theory by gathering lived experiences of successful management of peripartum emergencies” further triggers me: why not investigate/evaluate emergency cases that were handled unsuccessfully (or less successfully) – which I believe are sometimes referred to as ‘adverse events’? Were there any adverse events at all in this particular hospital? If yes, could one not learn more from a less successfully managed emergency case (even if only one) than from cases that were managed well?

• Methods: the topic of this study would lend itself perfectly for data collection methods other than, or at least complementary to individual interviews – for instance focus group discussion; or group model building (led by 2-3 moderators) with participants jointly developing a causal loop diagram. Has this been considered at all? I would have expected the authors to say something in the Study limitations about the method used in their study and possible alternative methods. The Study limitations subsection mentions “the use of additional data sources and triangulation” but it is not quite clear what this refers to.

• In the results section: “… teamwork also becomes the framework for negotiation of many conflicts that originated elsewhere.” This statement cries for more detail: what kind of conflicts did the researchers come across, how did they present themselves, how did they affect clinical management, did they (negatively) affect the outcome of the performed obstetric procedures?

• The Discussion section seems to be a little detached from the Findings. As I said, I’m not an expert in andragogy or related disciplines, but I do think that the discussion on power imbalances could have reflected a bit more on factors such as expertise, formal hierarchy, claims to moral authority, and epistemic versus normative power.

6. PLOS authors have the option to publish the peer review history of their article (what does this mean?). If published, this will include your full peer review and any attached files.

Reviewer #1: No

Reviewer #2: No

Reviewer #3: No

---

## [Author Response · Author response to Decision Letter 0]

28 Apr 2022

Dear Editor, dear reviewing Colleagues,

We would like to thank you for the opportunity to revise the manuscript, and we have attached both a clean version and one that tracked the changes made for further review purposes. We have addressed the reviewers’ comments in a separate document. 

We would like to extend our sincere thanks to all of you involved in the review process. Thanks to your critical and constructive remarks, you have uncovered important areas of improvement for the manuscript, and we hope to have adequately addressed them in our expanded version. 

Sincerely,

Ch.Neuhaus

---

## [Editor Report · Decision Letter 1]

27 May 2022

Emergence of Power and Complexity in Obstetric Teamwork

PONE-D-21-32827R1

Dear Dr. Neuhaus,

We’re pleased to inform you that your manuscript has been judged scientifically suitable for publication and will be formally accepted for publication once it meets all outstanding technical requirements.

Kind regards,

Dylan A Mordaunt, MD, MPH, FRACP

Academic Editor

PLOS ONE

Additional Editor Comments (optional):

Thank you for your resubmission. This now meets the criteria for publication.
---

## [Editor Report · Acceptance letter]

31 May 2022

PONE-D-21-32827R1 

Emergence of Power and Complexity in Obstetric Teamwork 

Dear Dr. Neuhaus:

I'm pleased to inform you that your manuscript has been deemed suitable for publication in PLOS ONE. Congratulations! Your manuscript is now with our production department. 

Kind regards, 

on behalf of

Associate Professor Dylan A Mordaunt 

Academic Editor

PLOS ONE